# Production and Physiological Quality of Seeds of Mini Watermelon Grown in Substrates with a Saline Nutrient Solution Prepared with Reject Brine

**DOI:** 10.3390/plants11192534

**Published:** 2022-09-27

**Authors:** Tatianne Raianne Costa Alves, Salvador Barros Torres, Emanoela Pereira de Paiva, Roseane Rodrigues de Oliveira, Renata Ramayane Torquato Oliveira, Afonso Luiz Almeida Freires, Kleane Targino Oliveira Pereira, Douglas Leite de Brito, Charline Zaratin Alves, Alek Sandro Dutra, Clarisse Pereira Benedito, Alberto Soares de Melo, Miguel Ferreira-Neto, Nildo da Silva Dias, Francisco Vanies da Silva Sá

**Affiliations:** 1Department of Agronomic and Forest Science, Federal Rural University of the Semi-Arid—UFERSA, Mossoró 59625-900, Brazil; 2Department of Agronomy, Federal University of Mato Grosso do Sul—UFMS, Chapadão do Sul 79560-000, Brazil; 3Department of Fitotecnia, Federal University of Ceará—UFC, Fortaleza 60356-001, Brazil; 4Department of Biology, State University of Paraíba—UEPB, Campina Grande 58429-500, Brazil

**Keywords:** *Citrullus lanatus*, germination, salinity, hydroponics, seed vigor

## Abstract

The economically profitable production of crops is related, among other factors, to seed quality, the production system, and the water used in irrigation or preparation of nutrient solutions. Therefore, the objective was to evaluate the phenology, production, and vigor of seeds of mini watermelons grown in saline nutrient solution and different substrates. In the fruit and seed production phase, the experiment occurred in a greenhouse with five electrical conductivities of water for nutrient solution preparation, ECw (0.5, 2.4, 4.0, 5.5, and 6.9 dS m^−1^), and two growing substrates (coconut fiber and sand). We evaluated the physiological quality of seeds previously produced under the five electrical conductivities of water and two substrates. High salinities for the hydroponic cultivation of the mini watermelon cultivar ‘Sugar Baby’ accelerated fruit maturation and crop cycle, decreasing fruit size. However, in both substrates, the seed production of mini watermelons, seed viability, and seed vigor occurred adequately with a reject brine of 6.9 dS m^−1^ in the hydroponic nutrient solution. The seed production of ‘Sugar Baby’ mini watermelons using reject brine in a hydroponic system with coconut fiber and sand substrates is viable in regions with water limitations.

## 1. Introduction

Low rainfall and high temperatures characterize the Brazilian semi-arid region for most of the year and high evapotranspiration rates [1]. Along with these factors, soil salinization is prevalent, causing losses in crop yield and quality [2].

Salinity is one of the main factors threatening agriculture and food security globally, mainly due to salts’ osmotic and ionic effects. These effects can interfere with cell stability and reduce plant water uptake, resulting in ion toxicity and changes in the cell’s physiological and metabolic processes [2,3]. Salt stress also alters hormone-regulated responses, limiting stomatal opening and photosynthesis [4]. These changes caused phenological changes resulting in reduced leaf expansion and photosynthetic area, lower plant growth rates, and smaller fruits and seeds [5].

Seeds’ physiological quality and vigor are essential for crop establishment [6]. Technical and environmental conditions of the production phase influenced the physiological seed quality [7], particularly stressful conditions such as salt stress. Excess salt can cause losses in the transport and accumulation of reserves, leading to low seed viability and vigor and an accelerated deterioration process [8].

The limited availability of good quality water for agriculture encourages strategies to use brackish water in agriculture. Brackish water desalination by reverse osmosis systems has guaranteed access to potable water for many communities in the Brazilian semi-arid region. However, water treatment by reverse osmosis generates brine waste, which must be used in fish farming or hydroponics to avoid soil contamination [5]. In this regard, using brackish water to prepare nutrient solutions for hydroponic crops is noteworthy as they do not present matric potential, and the roots are in a constant state of saturation, thus reducing the deleterious effects of salinity on the plants [9].

Agricultural producers are increasingly turning to hydroponic cultivation in protected environments, which enables intensive production and a continuous supply of products. Compared to conventional crops, the production cycle in hydroponics is shorter and more productive, and the products present a better quality [10]. In protected environment agriculture, hydroponics has been cultivated in substrates to provide the plant with excellent stability and support, supply oxygen, and promote carbon dioxide exchange between the roots and the external air [11]. Using coconut fiber substrate has demonstrated promising results in hydroponic cultivation with saline water [12,13].

Watermelon (*Citrullus lanatus* Schrad.), which belongs to the Cucurbitaceae family, is one species that thrives in hydroponic systems. It is one of the most important crops grown in Brazil. The Northeast region is the largest producer of watermelons in Brazil, accounting for 38% of national production [14]. In hydroponic cultivation, mini watermelon plants present great productive potential in all-year seasons, depending on management practices [15,16]. However, there are no studies on the effects of saline water in hydroponic cultivation of this species for phenology, production, and the physiological quality of seeds. This study aimed to evaluate the phenology, production, and vigor of seeds of mini watermelon grown in a nutrient solution prepared with reject brine and two growing substrates.

## 2. Results

During the mini watermelon cycle, there was a significant interaction (*p* < 0.05) between salinity and substrates. Single effect of salinity occurred only for the days to flowering—DTF (*p* < 0.001), and for the days from flowering to fruit maturity—DFFTFM (*p* < 0.001). There were also single effects of substrates on DTF (*p* < 0.001) and DFFTFM (*p* < 0.01). Mini watermelon flowering was delayed by eight days (Figure 1A) and there was a reduction of three days from flowering to fruit maturity (Figure 1C) for plants cultivated in the sand compared to those cultivated in coconut fiber.

Flowering was delayed by approximately one day for each unit increment in ECw, with a difference of 6.2 days when irrigation was performed with water of the highest salinity (6.9 dS m^−1^) compared to the control (0.5 dS m^−1^) (Figure 1B). The days from flowering to fruit maturity were reduced by 0.62 days for each increment of 1 dS m^−1^ in irrigation water, corresponding to a difference of 3.97 days in the treatment of the highest ECw (6.9 dS m^−1^) compared to the control (0.5 dS m^−1^) (Figure 1D).

The plant growth cycle in coconut fiber lasted an average of 81 days, regardless of salinity. The plant growth cycle in sand lasted from 83.6 to 88.6 days between salinity levels of 0.5 and 6.9 dS m^−1^, and the highest plant growth cycle occurred in the ECw of 4.9 dS m^−1^ (Figure 2). The cycles of plants cultivated in the sand were 2.6, 7.0, 7.0, 5.3, and 7.0 days longer than those grown in coconut fiber, at salinity levels of 0.5, 2.4, 4.0, 5.5, and 6.9 dS m^−1^, respectively (Figure 2).

The factors salinity, substrate, and time had significant interaction (*p* < 0.001) for longitudinal fruit diameter (LFD) and transverse fruit diameter (TFD) of a mini watermelon. In coconut fiber cultivation, the severe effects of salinity started five days after anthesis (DAA) for longitudinal and transverse diameters (Figure 3A,C). For longitudinal diameter, salinities S1 and S2 did not differ during the 45 DAA, but salinities S3, S4, and S5 differed from S1 and S2 from 20, 15, and 5 DAA, respectively (Figure 3A). At 45 days, the lowest longitudinal diameters occurred in salinity S5, with a reduction of 17.17%, followed by S4 and S3, with reductions of 4.37 and 7.74% compared to S1, respectively (Figure 3A). For transverse diameter, salinities S2, S3, S4, and S5 differed from S1 from 30, 15, 5, and 5 DAA, respectively (Figure 3C). At 45 days, reductions of 3.9, 5.6, 10.2, and 12.9% in the transverse diameter were seen under salinities S2, S3, S4, and S5 compared to S1, respectively (Figure 3C).

In sand cultivation, the effects of salinity began at 10 DAA for longitudinal and transverse diameters (Figure 3B,D). S1 was superior to the other salinities for longitudinal diameter on all days after anthesis, and S2 was superior to S3, S4, and S5 from 15 DAA (Figure 3B). At 45 days after anthesis, there were reductions of 2.35, 13.32, 10.44, and 10.83% in longitudinal diameter under salinities S2, S3, S4, and S5 compared to S1, respectively (Figure 3B). Salinities S3, S4, and S5 were similar for longitudinal diameter on all days after anthesis (Figure 3B). For transverse diameter, salinities S1 and S2 did not differ during the 45 DAA and were superior to S3, S4, and S5 from 15 DAA (Figure 3C). At 40 DAA, the transverse fruit diameters of plants under salinities S3, S4, and S5 decrease by 12.06, 7.56, and 7.10% compared to S1, respectively (Figure 3C). There was no significant difference in the transverse diameter of the fruits for those cultivated under salinities S4 and S5 (Figure 3C).

The interaction between salinity and growing substrates had a significant effect (*p* < 0.05) on fruit weight (FW) and hundred-seed weight (HSW). The substrate factor was significant for the weight of seeds per fruit (WSF) (*p* < 0.01) and for seed thickness (ST) (*p* < 0.001). The number of seeds per fruit (NSF), seed length (SL), and seed width (SW) were not significantly (*p* > 0.05) affected by the factors studied.

The increase in irrigation water salinity reduced the weight of mini watermelon fruits by 50.5 and 29.5% for plants grown in coconut fiber and sand substrates compared with those obtained at high (6.9 dS m^−1^) and low (0.5 dS m^−1^) salinity levels (Figure 4A).

The weight of mini watermelon fruits obtained under coconut fiber cultivation was higher than that obtained under sand cultivation for all salinities, precisely 75.9, 71.0, 75.6, 53.5, and 20.2% higher at the salinities of 0.5, 2.4, 4.0, 5.5, and 6.9 dS m^−1^, respectively (Figure 4A). The weight of mini watermelon seeds per fruit cultivated in coconut fiber was 60.5% (2.3 g) higher than that obtained for sand cultivation (Figure 4B).

The average hundred-seed weight of plants grown in the sand was 3.41 g, regardless of salinity. In coconut fiber cultivation, the hundred-seed weight ranged from 3.27 to 3.67 g between salinities of 0.5 and 6.9 dS m^−1^, and the estimated salinity of 2.45 dS m^−1^ obtained the highest value (Figure 4C). For the hundred-seed weight, the substrates did not differ at the salinity of 0.5 dS m^−1^. At salinities of 2.4 and 5.5 dS m^−1^, the hundred-seed weight of watermelon cultivated in coconut fiber was 14.2 and 10.0% higher than the values obtained with sand cultivation, respectively. At the salinity levels of 4.0 and 6.9 dS m^−1^, the hundred-seed weight of plants grown in the sand was 2.5 and 3.2% higher than those obtained with coconut fiber cultivation, respectively (Figure 4C).

The thickness of seeds of mini watermelon cultivated in coconut fiber was 7.0% (0.11 mm) higher than that obtained with sand cultivation (Figure 4D). Fruits produced in coconut fiber substrate obtained higher seed weight. Therefore, this result is not related to the number of seeds but their weight due to their greater thickness.

For the variables that indicate the physiological quality of the seeds, there was a significant interaction between the factors of salinity and substrates for germination (*p* < 0.05), electrical conductivity (*p* < 0.001), and accelerated aging (*p* < 0.01). The salinity factor significantly affected the emergence (*p* < 0.01).

The germination of mini watermelon seeds from coconut fiber cultivation was not influenced by water salinity, with an average of 100% (Figure 5A). Seeds from sand cultivation obtained the highest germination (100%) at a salinity of 2.9 dS m^−1^ (Figure 5A). There was a difference between the substrates under a salinity of 6.9 dS m^−1^, and seeds from coconut fiber obtained 19 percentage points more than seeds from sand cultivation (Figure 5A).

The highest percentage of seedling emergence (87%) was obtained in the treatment with ECw 5.0 dS m^−1^, being 17% above the control (0.5 dS m^−1^) regardless of the substrate used (Figure 5B). Irrigation with saline water increased the electrical conductivity test of mini watermelon seeds by 3.84 and 3.19 μS m^−1^ for seeds of watermelon grown in coconut fiber and sand, respectively, for each increase of 1 dS m^−1^ in irrigation water (Figure 5C). The electrical conductivity test of mini watermelon seeds produced in a sand substrate was 8.6, 11.0, 10.3, and 20.6 μS m^−1^ higher than that of seeds produced in coconut fiber for salinities of 0.5, 2.4, 4.0, and 5.5 dS m^−1^, respectively (Figure 5C). At a salinity of 6.9 dS m^−1^, there was no significant difference between substrates for the electrical conductivity test of mini watermelon seeds.

The mini watermelon seeds obtained in coconut fiber cultivation after accelerated aging were not influenced by the irrigation water salinities, obtaining an average of 89% (Figure 5D). In turn, seeds produced in the sand obtained the highest germination after accelerated aging (97%) under a salinity of 3.74 dS m^−1^ (Figure 5D). There was a difference between the substrates under salinity of 4.0 dS m^−1^, at which the seeds produced in sand obtained levels 14% above those produced with coconut fiber cultivation (Figure 5D).

## 3. Discussion

The reuse of agro-industrial products in agriculture is essential for clean production. We produce mini watermelon fruits and seeds using brine waste from water desalination in rural communities and coconut fiber from residue from coconut water production.

The mini watermelon plant’s cycle differed with the variations in the electrical conductivity (EC) of the water and cultivation substrates. The cycle of the plants grown in the coconut fiber substrate had an average duration of 81 days, whereas sand cultivation prolonged the cycle by 2.6 to 7 days. Watermelon plants cultivated in sand grew faster and reached the maximum height of the trellis (2.0 m). However, the plants needed more time to reach the top of the trellis when a nutrient solution containing water with a high concentration of salts was used. Thus, plants grown in the sand under higher salt stress presented longer cycles. Although salinity associated with sand cultivation reduced fruit maturation time, the overall length of the crop cycle did not decrease.

The physiological effects on the mini watermelon plants grown in sand and under salt stress prolonged the crop cycle. Drought induced by the reduction in osmotic potential due to the increased salinity of the water used in cultivation was the main factor that caused the mini watermelon plant’s reduced growth [15,17,18]. Due to this and the sand’s low water retention capacity, the plant growth rate decreased; consequently, the main branches needed more time to grow to the proper length.

The longitudinal and transversal diameters of the mini watermelons grown in the coconut fiber substrate at 45 days after anthesis (DAA) were 12.1% and 10.5% larger, respectively, compared with the diameters of the mini watermelons grown in the sand. The longitudinal and transversal diameters of the mini watermelons cultivated in coconut fiber were superior to those of the mini watermelons grown in the sand at 15 DAA for salinity treatments S1, S2, S3, S4, and S5. The longitudinal and transversal diameters resulting from coconut fiber cultivation under salinity treatments S1 and S2 were similar and, along with treatment S3, were superior to the diameters resulting from sand cultivation under treatment S1. The results from cultivation in the coconut fiber substrate under salinity treatments S4 and S5 were similar to those of sand cultivation under treatments S1 and S2, respectively. Therefore, the coconut fiber substrate promoted fruit growth and reduced the effect of salinity on the mini watermelon plants compared to sand. Coconut fiber has a greater capacity for hydration and water retention than sand without restricting aeration; as a result, coconut fiber makes more water available to the plant, even under conditions of low osmotic potential [19].

The fruits of the plants grown in the coconut fiber substrate presented greater diameters and fruit weight than those grown in sand, regardless of water salinity. The extended period required for fruit maturation and larger diameters significantly increased fruit weight. Fruit weight was 75% higher in plants grown in coconut fiber than in the sand under salinity treatment S1. For EC above 4.0 dS m^−1^, the difference between the fruit weight of the crops grown in coconut fiber and those grown in sand decreased, which indicates that salt stress was more limiting in the coconut fiber substrate at this salinity level. Severe restrictions in fruit weight with a salinity condition of 4.0 dS·m^−1^ corroborate the reduction in the fruit’s maturation period and longitudinal and transversal diameters.

Thus, decreases in fruit size and weight were not due to poor fertilization and poor fruit formation, which are capable of reducing sink strength (fruit) [20]. However, the restricted availability of water and photoassimilates due to the osmotic and ionic salinity components limits root development, leaf growth and expansion (source), and water relations and photosynthesis [2,3,4].

The mini watermelons grown in coconut fiber presented similar diameters and weights to those reported in the literature. However, the results we obtained for the plants grown in the sand substrate are lower than those reported by [4,15,16] for the hydroponic cultivation of the ‘Smile’ and ‘Sugar Baby’ cultivars of the mini watermelon using saline water in the nutrient solution. The authors of [21] observed mini watermelons under salinity conditions of 2.0 and 5.2 dS·m^−1^ and found that an increase in salinity resulted in a 240 g reduction in fruit weight for the non-grafted ‘Tex’ cultivar, 17.4 g of which corresponded to a reduction in seed weight. In the ‘Sugar Baby’ cultivar, fruit weight reduction between salinities of 2.4 and 5.5 dS·m^−1^ was similar to that observed in our study. However, we did not detect seed weight per fruit reduction due to the water’s increased salinity, but mini watermelons grown in coconut fiber obtained more seed weight per fruit and thickness than seeds produced in sand cultivation. The higher water retention capacity of coconut fiber compared to the area favored the production of mini watermelon seeds with brackish water.

Despite the influences on the phenology and production of mini watermelon fruits and seeds grown in a hydroponic system with brackish water, seed viability in this system was acceptable. All treatments obtained germination over 90%, except for seeds from sand cultivation irrigated water of 6.9 dS·m^−1^, which presented 84% of germination and seedling emergence between 70% and 87%. The percentages we found for germination and seedling emergence are within the appropriate range for the watermelon crop [22,23], which indicates that cultivation with brackish water did not affect the seed viability of the ‘Sugar Baby’ mini watermelon.

The seeds produced under higher salinity conditions presented greater EC values in the leaching water compared to the control (0.5 dS·m^−1^), mainly when cultivated in the sand substrate. The metabolic disturbances caused by osmotic [18] and ionic [3] components during seed formation damaged the membranes, causing more significant electrolyte extravasation [7]. Electrolyte extravasation was more evident in the plants grown in the sand, in which production was affected more by salt stress than in the plants grown in coconut fiber. Although the EC values of the leaching water of the seeds rose, germination in the accelerated aging test was above 80% in all treatments, which indicates high seed vigor [23]. Although salt stress caused some damage during the production phase of ‘Sugar Baby’ mini watermelon fruits and seeds, the seeds showed good viability and vigor. Both [24] and [25] showed that saline stress decreases ‘Sugar Baby’ mini watermelon growth, fruit production, and post-harvest quality. They showed that salt stress does not affect the mini watermelon photosynthetic rate due to its high photosystem II efficiency. However, salinities from 4 dS m^−1^ significantly decrease plant growth, fruit production, and fruit post-harvest quality. They obtained marketable quality fruits only in mini watermelons grown in coconut fiber and irrigated with brackish water of up to 4.0 dS m^−1^. They did not evaluate the salinity effect on phenology, seed yield, and seed quality. We considered it; we found that saline stress alters the phenology of mini watermelon, decreasing the cycle length and time from flowering to fruit maturation. We verified that the decrease in fruit cycle, size, and weight occurred in both substrates; however, the decrease for every 1 dS m^−1^ increase in water salinity in the mini watermelon grown on coconut fiber exceeded those obtained with sand substrate. However, we find that the seed production with good physiological quality in the coconut fiber and sand substrates occurred up to 6.9 dS m^−1^. We found that changes in watermelon phenology by salinity decreased fruit production but did not impair seed production. The present research results reinforce the sustainable management of brackish water by the indicators such as the weight of seeds per fruit, hundred-seed weight, germination, emergence, seed electrical conductivity test, and seed accelerated aging. Therefore, our research makes it possible to identify further the potential use of reject brine in the hydroponic cultivation of mini watermelon in substrates. Thus, the seed production of mini watermelon with reject brine in the hydroponic cultivation is an alternative for regions with little available water.

## 4. Materials and Methods

### 4.1. Location and Characterization of the Environment

The study was conducted at the Federal Rural University of the Semi-Arid Region (UFERSA), Mossoró, RN, Brazil, and consisted of two phases. The first was carried out in a greenhouse (phenology and seed production) and the second in the laboratory (physiological quality of mini watermelon seeds produced in the previous phase). During the experiment in the greenhouse, the maximum and minimum values recorded in the environment were 39.2 and 20.4 °C for temperature and 86 and 22% for relative humidity, respectively.

### 4.2. Phase I—Phenology and Seed Production

The treatments were distributed in a split-plot scheme with a randomized block design (RBD). The plot was composed of the five electrical conductivities of water for nutrient solution preparation, ECw (S1 = 0.5, S2 = 2.4, S3 = 4.0, S4 = 5.5, and S5 = 6.9 dS m^−1^). The subplot was composed of two substrates (coconut fiber and sand), with four replicates of two plants.

The cultivar used was ‘Sugar Baby’, which has a rounded shape, dark green rind, bright red flesh, and few seeds. Cultivation was performed in 6-dm^3^ plastic pots filled with the growing substrates. The coconut fiber substrate has a fine texture, 95% total porosity, 507 ml L^−1^ (substrate) of water retention capacity, 0.5 dS m^−1^ (EC 1:5) of electrical conductivity, and 6.0 pH (pH 1:5). The sand substrate was sieved through a 4 mm mesh and washed with tap water. Three seeds were sown in each hole, and thinning was performed on the fifth day after sowing, leaving only one plant. Mini watermelon plants were trained in a vertical trellis with 2.0 m height, in five rows with 1.00 m spacing, with 16 plants in each row spaced 0.30 cm apart. During the growth, excess lateral shoots were eliminated up to the ninth branch by pruning, leaving the other shoots with five leaves. The apical bud was eliminated when the plants reached 2 m in height, leaving only one fruit per plant. Pollination was carried out manually during the early morning hours, and the fruits were placed in plastic baskets.

Until the 10th day of cultivation, irrigation was performed with water from the supply network (EC = 0.54 dS m^−1^). After this period, the nutrient solutions prepared with the different salinities began to be applied. The saline waters were obtained by mixing the public-supply water (PSW) and reject brine water from desalination (RBW), in the following proportions: S1—100% PSW, S2—85% PSW + 15% RBW, S3—70% PSW + 30% RBW; S4—55% PSW + 45% RBW; S5—40% PSW + 60% RBW. The public-supply water showed the following chemical composition: pH = 7.57; ECw = 0.5 dS m^−1^; Ca^2+^ = 0.83 mmolc L^−1^; Mg^2+^ = 1.20 mmolc L^−1^; K^+^ = 0.31 mmolc L^−1^; Na^+^ = 3.79 mmolc L^−1^; Cl^−^ = 2.40 mmolc L^−1^; CO_3_^2−^ = 0.60 mmolc L^−1^; HCO^3−^ = 3.20 mmolc L^−1^; and SAR = 3.76 (mmolc L^−1^)^−0.5^. Reject brine was collected in the Jurema Rural Settlement, Tibau, RN, Brazil, with the following chemical composition: pH = 7.10; ECw = 9.5 dS m^−1^; Ca^2+^ = 37.8 mmolc L^−1^; Mg^2+^ = 24.20 mmolc L^−1^; K^+^ = 0.83 mmolc L^−1^; Na^+^ = 54.13 mmolc L^−1^; Cl^−^ = 116.00 mmolc L^−1^; CO_3_^2−^ = 0.00 mmolc L^−1^; HCO^3−^ = 3.40 mmolc L^−1^; and SAR = 9.70 (mmolc L^−1^)^−0.5^.

The nutrient solutions were applied twice a day, in the early morning and the late afternoon, considering the volume corresponding to the actual evapotranspiration of the crop, measured by drainage lysimeters in additional plots corresponding to each treatment. A drip irrigation system applied the depth, composed of 16-mm-diameter hoses and pressure-compensating drippers with a flow rate of 1.4 L h^−1^, connected to a self-venting Metalcorte/Eberle circulation motor pump, driven by a single-phase motor, 210 V voltage, 60 Hz frequency, installed in a reservoir with 50 L capacity.

The standard nutrient solution proposed by [26] was used for macronutrients. Micronutrients were supplied using the commercial compound Rexolin BRA, which consists of 11.68% potassium oxide (K_2_O), 1.28% sulfur (S), 2.1% boron (B), 0.36% copper (Cu), 2.65% iron (Fe), 2.48% manganese (Mn), 0.036% molybdenum (Mo), and 3.38% zinc (Zn), following the manufacturer’s recommendation (2 g L^−1^). The nutrient solution has an electrical conductivity of 1.1 dS m^−1^, and after preparation the solutions showed the following electrical conductivities: 1.6, 3.5, 5.1, 6.6, and 8.0 dS m^−1^.

The experiment evaluated plants for phenology and fruit and seed production. The phenological variables considered were: days to flowering (DTF), by counting the days from sowing to the emergence of the open flower, considering the beginning of flowering as the moment when 50% of the plants in the treatment had at least one open flower per plant; days from flowering to fruit maturity (DFFTFM), by counting the days from anthesis (flower opening) to the physiological maturity of the fruit (harvest point), considering female flowers that had an ovary with a transverse diameter of 2 cm as fruits, while the harvest point of the fruits was defined based on the guidelines of [12], who consider fruits with completely dry tendril coming from the same node; and cycle length, by counting the days from sowing to fruit harvest.

The variables related to fruit production were: transverse fruit diameter (TFD) and longitudinal fruit diameter (LFD), measured with a digital caliper at 5, 10, 15, 20, 25, 30, 35, 40, and 45 days after anthesis, with results expressed in millimeters; and fruit weight (FW), determined by manual harvesting of the fruits, followed by a weighing on an analytical scale, with results expressed in grams.

After harvesting and weighing the fruits, the mini watermelon seeds were extracted manually with a spoon and a sieve, washed, and dried naturally to remove the mucilage. These seeds were then evaluated for the following variables: the number of seeds per fruit (NSF) by manually counting the seeds produced in each fruit; the weight of seeds per fruit (WSF) by weighing the fresh seeds extracted from each fruit on a precision analytical scale, with results expressed in grams; hundred-seed weight (HSW), for which eight replicates of 100 fresh seeds of each treatment were separated and subsequently weighed on a precision analytical scale, with results expressed in grams; and seed length (SL), width (SW), and thickness (ST), measured with a digital caliper using 10 seeds per treatment, with results expressed in millimeters.

### 4.3. Phase II—Physiological Quality of Seeds

The viability and vigor of the mini watermelon seeds produced in the first phase were evaluated in a completely randomized delineation (CRD), with four replicates of 50 seeds. The seeds produced under five electrical conductivities of irrigation water, ECw (S1 = 0.5, S2 = 2.4, S3 = 4.0, S4 = 5.5, and S5 = 6.9 dS m^−1^), and two substrates (coconut fiber and sand) were considered as lots. For this, the seeds were extracted from the fruit manually, washed in running water to remove the mucilage, and dried in the natural environment (30 °C) for 24 h. Then, their initial moisture content was determined using the oven method at 105 ± 3 °C, with two replicates of 4 ± 0.05 g for 24 h [27]. The results were expressed as a percentage (wet basis) (Table 1).

The germination test was conducted in a Biochemical Oxygen Demand (B.O.D.)-type germination chamber, at 25 °C, with a photoperiod of eight hours and in paper towel roll substrate moistened with distilled water in an amount equivalent to 2.5 times the dry weight. Normal seedlings were counted 14 days after sowing [27].

The emergence test was performed in a greenhouse, using four replicates of 50 seeds. Sowing was carried out in polyethylene trays containing the coconut fiber substrate and irrigated with public-supply water. Emerged seedlings were counted 14 days after sowing, and the emergence percentage was calculated later.

The mass method conducted the electrical conductivity test with four replicates of 50 seeds, which were weighed, placed in plastic cups containing 75 mL of distilled water, and kept at a constant temperature of 25 °C for 24 h of incubation. After this period, the electrical conductivity of the solution was determined in a Digimed CD-21 conductivity meter, and the results were expressed in μS^−1^ cm^−1^ g^−1^ of seeds [28].

To conduct the accelerated aging test, a single layer of seeds was placed on a metal screen fixed in an acrylic box containing 40 mL of distilled water. The closed containers were kept in germination chambers (B. O. D.) at 41 °C for 48 h [29]. After this period, the germination test evaluated four subsamples of 50 seeds, computing the percentage of normal seedlings five days after sowing.

### 4.4. Statistical Analysis

The data were subjected to analysis of variance by the F test, and the Tukey analyzed the effects of the treatments means comparison test at a 5% probability level and polynomial regression analysis. Statistical analyses were performed using the statistical software Sisvar 5.7 [30].

## 5. Conclusions

High salinities for the hydroponic cultivation of the mini watermelon cultivar ‘Sugar Baby’ accelerated fruit maturation and crop cycle, decreasing fruit size. The greater cycle acceleration in plants grown in coconut fiber caused a marked reduction in fruit weight compared to plants grown in sand. However, at all salinities, the fruits of plants grown on coconut fiber outperform plants grown on sand. The fruits of plants grown in sand and those grown in coconut fiber with 6.9 dS m^−1^ were inferior in size and weight. However, in both substrates, the seed production of mini watermelon, seed viability, and seed vigor occurred adequately with a reject brine of 6.9 dS m^−1^ in the hydroponic nutrient solution. We found that salt stress affects the fruit production of mini watermelon, but was not harmful for seed production. The seed production of ‘Sugar Baby’ mini watermelons using reject brine in a hydroponic system with coconut fiber and sand substrates is viable in regions with water limitations.

## Figures and Tables

**Figure 1 plants-11-02534-f001:**
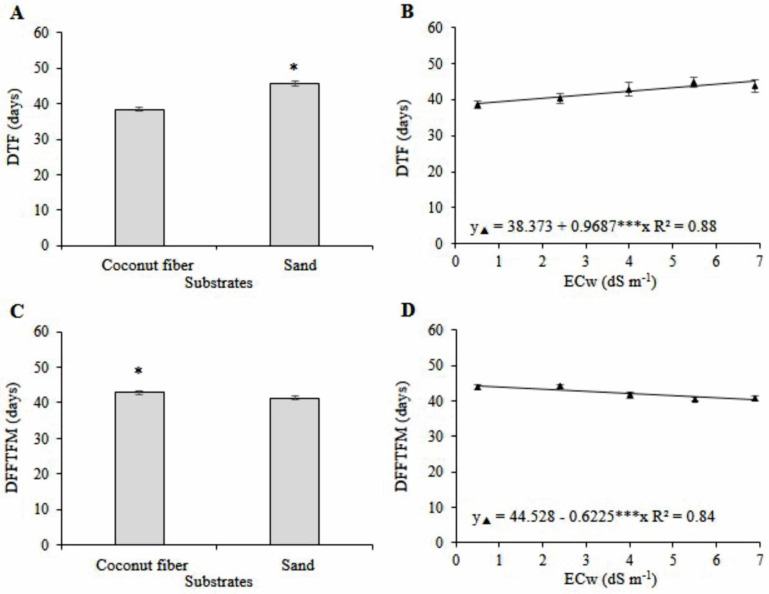
Regression analysis and means comparison test (Tukey, *p* < 0.05 and SD, *n* = 20) for the variables days to flowering—DTF (**A**,**B**) and days from flowering to fruit maturity—DFFTFM (**C**,**D**) of mini watermelon in hydroponic cultivation using a nutrient solution prepared with reject brine and different substrates. *** and * significant at 0.001 and 0.05 probability levels, respectively.

**Figure 2 plants-11-02534-f002:**
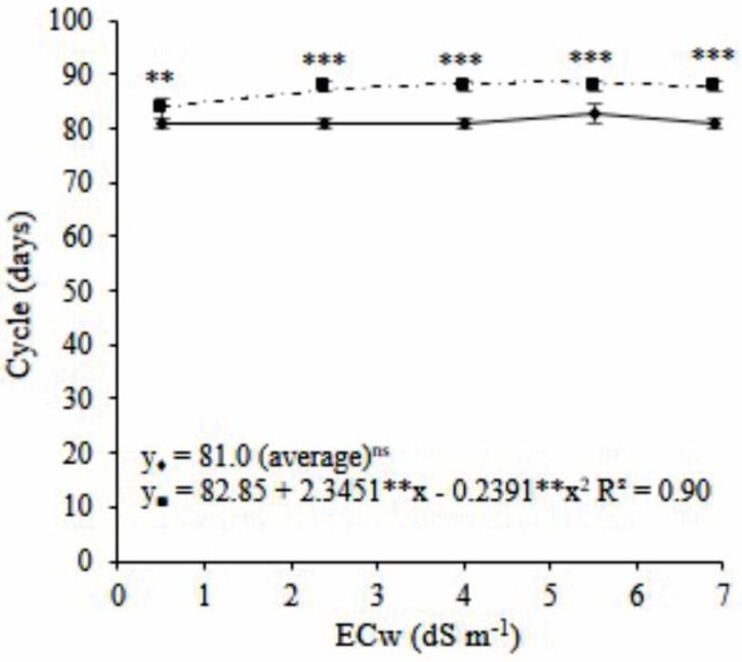
Regression analysis and means comparison test (Tukey, *p* < 0.05 and SD, *n* = 4) for the cycle of mini watermelon in hydroponic cultivation using a nutrient solution prepared with reject brine and different substrates. (♦ Coconut fiber; ■ Sand). ***, **, and ns significant at 0.001, 0.01 probability levels, and not significant, respectively.

**Figure 3 plants-11-02534-f003:**
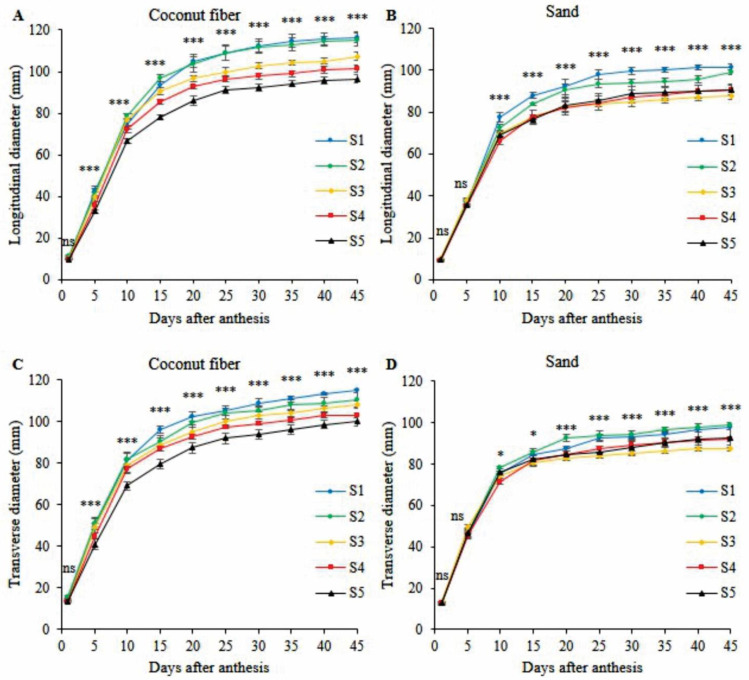
Means comparison test (Tukey, *p* < 0.05 and SD, *n* = 4) for longitudinal diameter (**A**,**B**) and transverse diameter (**C**,**D**) of mini watermelon fruits as a function of days after anthesis, in hydroponic cultivation using nutrient solution prepared with reject brine (S1—0.5 dS m^−1^, S2—2.4 dS m^−1^, S3—4.0 dS m^−1^, S4—5.5 dS m^−1^, S5—6.9 dS m^−1^) and substrates (Coconut fiber and Sand). ***, * and ns significant at 0.001, 0.05 probability levels, and non-significant, respectively.

**Figure 4 plants-11-02534-f004:**
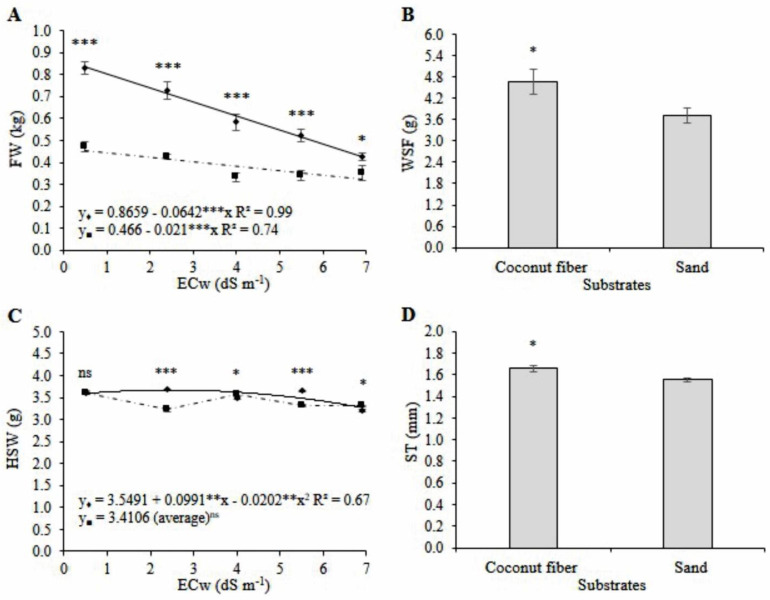
Regression analysis and means comparison test (Tukey, *p* < 0.05 and SD, *n* = 20) for the variables fruit weight–FW (**A**), the weight of seeds per fruit–WSF (**B**), hundred-seed weight–HSW (**C**), and seed thickness –ST (**D**) of mini watermelon in hydroponic cultivation using a nutrient solution prepared with reject brine and different substrates (♦ Coconut fiber; ■ Sand). ***, **, *, and ns significant at 0.001, 0.01, 0.05 probability levels, and not significant, respectively.

**Figure 5 plants-11-02534-f005:**
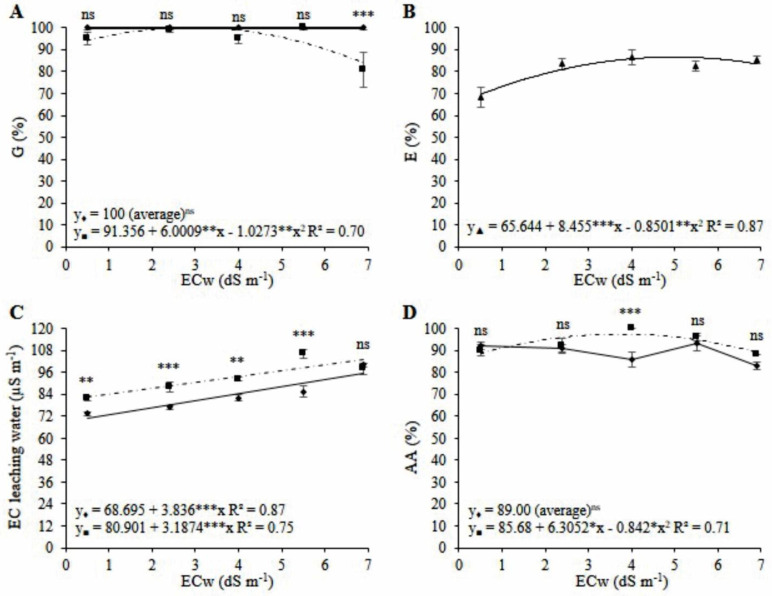
Regression analysis and means comparison test (Tukey, *p* < 0.05 and SD, *n* = 4) for the variables germination—G (**A**), emergence—E (**B**), electrical conductivity test—EC leaching water (**C**), and accelerated aging—AA (**D**) of seeds of mini watermelon in hydroponic cultivation using a nutrient solution prepared with reject brine and different substrates (♦ Coconut fiber; ■ Sand). ***, **, *, and ns significant at 0.001, 0.01, 0.05 probability levels, and not significant, respectively.

**Table 1 plants-11-02534-t001:** The initial moisture content of mini watermelon seeds is produced with reject brine (ECw) and substrates.

Seed Moisture Content (%)		
ECw (dS m^−1^)	Substrates	
	Sand	Coconut fiber
S1—0.5	8.4	8.4
S2—2.4	8.4	8.3
S3—4.0	8.4	7.7
S4—5.5	8.4	7.9
S5—6.9	8.7	8.6

## Data Availability

All other data are presented in the paper.

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
