# Peer review of "Production and Physiological Quality of Seeds of Mini Watermelon Grown in Substrates with a Saline Nutrient Solution Prepared with Reject Brine"

_plants, 2022, doi:10.3390/plants11192534_

Round 1

Reviewer 1 Report

This study was original and well-performed to the experimental design

There are some issues that need to improve the manuscript

1. The abbreviation of DTFM should be explained in detail so that the reader can understand it easily just by looking at the picture title. For example, "DTFM is from flowering to fruit maturity."

2. in fig. SD means standard deviation ? not error ?

3. fig. 1. title : days to flowering - DTF ---> days to flowering (DTF) 

    this format can be applied to the titles of other pictures as well.

4. The picture has a low resolution. The resolution is clear and needs correction.

5. In Figure 2, y = 81.0ns ? Does this have no formula ?

6. In Figure 5, EC would like to add a description to the title leaching water.

7. What was the EC of the coconut fiber used in this experiment? Many experiments have reported that coconut fiber itself has salt.

8. did the authors see some kind of differences in seedling growth in this experiment ?

9. In this experiment, WSF and ST differed between treatments (substrates). It would be nice to add consideration from a physiological point of view as to why this is the case.

Author Response

Dear Reviewer

  1. We corrected the abbreviation of DTFM to DFFTFM.
  2. The standard error is the product of the standard deviation divided by the square root of the number. So it's like displaying "SD, n= x" to indicate how standard error is done.
  3. We added images with 500 dpi.
  4. The "y = 81.0ns" means the mean, as there was no regression adjustment. We adjust.
  5. We have added “EC leaching water” in Figure 5C.
  6. We added the coconut fiber information in the material and methods.
  7. We did not measure plant growth but only allowed female flowers to fertilize when the branch length reached the top of the trellis.
  8. We've added the discussion about WSF and ST.

We have highlighted the changes you and other reviewers suggested in red in the text.

We thank you for reviewing and recommending our article for publication.

Best regards,

Francisco Vanies da Silva Sá, D.Sc. & Professor

Reviewer 2 Report

In the manuscript entitled: "Production and physiological quality of seeds of mini watermelon grown in substrates with a saline nutrient solution prepared with reject brine" (Number: plants-1909130), the Authors focused on the topic of cultivating an industrially important plant in hydroponics. As it is stated, salinity is a serious problem for efficient agricultural production, and therefore the presented topic is interesting and follow current interest in agriculture.

However, the Authors should explain more in the Introduction section (not only in the Discussion) why they use brackish water (form recycling) for their experiment/cultivation. Since, in hydroponic conditions, thus under controlled conditions, we can provide plants with water having highly controlled conditions, without pollution; so please explain more about the need to use brackish water in this particular experiment. The serious issue represent the control experiment, the presented research lacks a control culture, i.e. germination-growth-fruiting on a clean medium - without salinity, it is critical; thus, the manuscript must be extended with additional analysis showing control experiment for comparison.

In conclusion, in my opinion, the scientific value of the research presented in the manuscript is low without proper control experiment, on the other hand, however detailed analyzes of watermelon growth under salinity conditions are of great value for agriculture.

Author Response

Dear Reviewer,

We added in the Introduction the importance of water recycling and the use of waste in hydroponics.

We have control. Water with an EC of 0.5 dS m-1 is tap water. It has a low risk of soil salinization and sodification.

We have highlighted the changes you and other reviewers suggested in red in the text.

We thank you for reviewing and recommending our article for publication.

Best regards,

Francisco Vanies da Silva Sá, D.Sc. & Professor
